# ECG-Based Stress Detection and Productivity Factors Monitoring: The Real-Time Production Factory System

**DOI:** 10.3390/s23125502

**Published:** 2023-06-11

**Authors:** Mssimiliano Donati, Martina Olivelli, Romano Giovannini, Luca Fanucci

**Affiliations:** 1Department of Information Engineering, University of Pisa, 56122 Pisa, Italy; martina.olivelli@phd.unipi.it (M.O.);; 2Digital Building srl, 00156 Rome, Italy

**Keywords:** Industry 4.0, work stress, machine learning, CNN, ECG, sensor network, production quality, worker well-being, productivity

## Abstract

Productivity and production quality have become primary goals for the success of companies in all industrial and manufacturing sectors. Performance in terms of productivity is influenced by several factors including machinery efficiency, work environment and safety conditions, production processes organization, and aspects related to workers’ behavior (human factors). In particular, work-related stress is among the human factors that are most impactful and difficult to capture. Thus, optimizing productivity and quality in an effective way requires considering all these factors simultaneously. The proposed system aims to detect workers’ stress and fatigue in real time using wearable sensors and machine learning techniques and also integrate all data regarding the monitoring of production processes and the work environment into a single platform. This allows comprehensive multidimensional data analysis and correlation research, enabling organizations to improve productivity through appropriate work environments and sustainable processes for workers. The on-field trial demonstrated the technical and operational feasibility of the system, its high degree of usability, and the ability to detect stress from ECG signals exploiting a 1D Convolutional Neural Network (accuracy 88.4%, F1-score 0.90).

## 1. Introduction

Productivity and production quality are among the key factors for the competitiveness of companies in all industrial and manufacturing sectors. In particular, productivity refers to the efficiency of goods production and is related to the quantity and quality of items produced in a certain period with respect to the resources (i.e., materials, equipment, labor) used during the process [1]. Therefore, productivity improvements do not mean increasing production rates while sacrificing the quality and reliability of the final products.

Increasing productivity requires the optimization of both operational and managerial processes within the production system [2], identifying bottlenecks, and minimizing the use of resources to obtain the final products at the expected quality level. In addition, it is important to decrease failures and defects due to human mistakes or equipment errors during production [3]. Such optimizations produce benefits in terms of production time and cost, enabling the company not only to have higher marginality on products sold but also to improve its strategic positioning on the market.

Several factors are recognized in the literature as affecting the productivity and quality of the final products in different production sectors [4,5,6,7,8], especially in the case of non-repairable products [9]. They can be grouped into three main macro categories: technical aspects, workplace and work environment, and organizational and human factors. Thus, productivity is a multidimensional concept, characterized also by mutual influences among the aforementioned factors, and in order to make improvements, it is important to take them into account simultaneously.

Technological factors include mainly the availability of efficient and precise plants, machinery and equipment involved in the production cycle, but also the choice of materials to be used during production. Regular maintenance and upgrading operations are fundamental to reduce defects in production. From a technical point of view, the advancement of production processes benefits from technological innovation [10], digitalization strategies [11,12], and more generally from the adoption of Industry 4.0 models [13]. The implementation of such models is based on Industrial Internet-of-Things (IIoT): digital technologies [14] that allow monitoring the performance and control of plants, machinery, industrial automation, and production flows in real time [15]. The resulting data enable to analyze and optimize processes, to prevent malfunctions, to improve the quality of the final products and also to increase the safety of workers.

Productivity is also affected by environmental conditions such as illuminance, temperature, relative humidity, air quality and noise [16]. In particular, an uncomfortable working environment can have a negative impact on workers’ performance and health [17] and therefore reduce job satisfaction [18]. Safety conditions within the workplace and ergonomics are equally important. The monitoring of the working environment conditions with specific sensing technologies allows for an additional important source of data to be considered for improving productivity, safety, and worker satisfaction.

Human factors are related to the nature and conduct of persons, and they are mainly concerned with the ability and willingness to perform the assigned tasks (e.g., personal skill, motivation and commitment, relationships with colleagues, well-being, etc.). They have an important impact on productivity and quality outcomes [19], and require an appropriate management strategy [20]. Among these factors, work stress has a great negative impact on personal and overall performance [21], leading to increases in the human error rate [19]. It also exposes the worker to potential risks of incidents and health problems [22,23,24]. Such stress is defined as an adaptive response to a situation that is perceived as challenging or threatening to personal well-being. The causes can be work overload, time pressure, demands beyond the worker’s ability, unclear responsibility and task assignments, lack of breaks or poor environmental conditions [25]. Three kinds of stress exist: acute when caused by short-term factors, episodic when it occurs periodically, and chronic when caused by long-term factors. The latter is the most harmful, as it can lead to serious health problems. In fact, stress produces an alteration of physiological parameters such as body temperature, heart rate, blood pressure, sugar level and their variability [26]. Moreover, it influences the behavior favoring anxiety, absenteeism, disputes, isolation, unsatisfaction, etc. To date, the monitoring of measurable human factors such as stress is not automated as for those related to the machinery, workplace and processes previously described. Conversely, the assessment is based on traditional methodologies, including periodic interviews with specialized physicians, dedicated questionnaires [27], and the detection of sentinel events (e.g., occupational accidents, disputes, absenteeism, etc.). This reactive method allows triggering corrective actions, but it is characterized by a poor propensity for prevention and early identification of stressful situations.

In order to monitor all factors affecting productivity and safety, it is is necessary to consider a large amount of data coming from different sources in an integrated way. For this purpose, this paper presents the Real-Time Production Factory System (RT-PROFASY) developed and tested under a research project co-funded by the Italian Ministry of Economic Development. The implemented system achieves two important goals within a single platform: first, the active monitoring of the worker’s stress level, fatigue, and health status during working activities; and secondly, the integration of stress-related data with data flows coming from production processes, factory environment, and indoor localization. This enables the elaboration and analysis of all relevant data, which are otherwise difficult to gather from separated systems and traditional stress detection methods. The final aim of this work is to provide a complete tool, with hardware and software components, to help companies to pursue productivity and product quality through sustainable processes for workers and appropriate workplace conditions, moving from a reactive to a proactive and fast monitoring method of worker status.

The RT-PROFASY system was proposed in [28]. This work provides additional details about the implemented system modules, the data elaboration algorithms, and the selection of Commercial Off-the-Shelf (COTS) hardware components. Moreover, it includes an in-depth presentation and discussion of the results achieved in the on-field trial. Hereafter, Section 2 analyzes the state of the art related to monitoring and analysis of factors affecting productivity and safety in the workplace and work stress detection. Section 3 presents the RT-PROFASY system in detail. The results of the on-field trial are discussed in Section 4, and finally, conclusions are drawn in Section 5.

## 2. State of the Art

The monitoring of machinery, equipment and plants involved in production processes is now widespread, especially in organizations with a high degree of automation. The main indicators of machinery operations in terms of performance, conditions and safety can be measured in real time using relatively inexpensive sensor networks, distributed data loggers (often embedded directly into the machines) and other IIoT-based systems easily deployable in existing plants [29,30,31]. In that respect, well-established monitoring architectures and strategies are presented in the literature [15,32]. The collected data provide a thorough understanding of the behavior of manufacturing processes and the possibility of implementing large-scale to fine-grained optimizations of the production cycles. They also enable the prevention of failures according to the recently developed predictive maintenance strategies [33]. In addition, IIoT solutions and modern technologies allow machine-to-machine communications and the active monitoring of human–machine interactions for safety purposes.

Sensor networks are also commonly used to monitor indoor environmental conditions in company facilities. The sensing devices acquire data from the surrounding environment and allow measuring parameters such as temperature, humidity, brightness, noise level, CO_2_ level, air quality [34,35,36], and other factors that impact worker productivity or occupant safety [37]. The collected data can then be used to create a more sustainable, efficient, and safe workplace.

The indoor localization and tracking of assets are other important aspects to consider for improving the organization and speed of workflows as well as the safety of operators. In particular, as far as safety is concerned, the possibility of localizing Personal Protection Equipment (PPE) within the working space or specific subregions enables the detection of PPE compliance vs. the activity performed by the worker [38]. Among the wide range of technologies and techniques suitable for localization purposes [39], Bluetooth Low Energy (BLE) beaconing [40,41], radio frequency identification based on passive tags (RFID) [42], and Ultra-Wide Band (UWB) [43] are the most promising. Despite this, their wide deployment is still rather limited and mainly concentrated in large companies and in sectors such as warehousing and logistics.

The detection of work-related stress conditions is normally carried out with offline methodologies (i.e., questionnaires, interviews, periodic medical examinations, etc.). While this method shows poor propensity for prevention and limitations in identifying acute stress situations, there is no evidence of widespread systems for the real-time monitoring of the stress level of the worker. There are instead in the literature interesting results of research projects focused on the detection and classification of stress and more in general emotions felt by a person subjected to stressful stimuli such as working under time-pressure conditions, continuous interruptions during activities, or viewing emotionally engaging images/videos during the experiment. These studies are mostly related to office-type environments or based on ad hoc laboratory setups that include video cameras, electromyographic helmets, multi-leads electrocardiogram devices and other devices for vital signs acquisition, which are difficult to apply during heavily dynamic activities typical of manufacturing companies [44,45,46,47,48]. Such types of working activities also impact the quality of signals measured by wearable devices, and the high level of noise with respect to (semi)stationary acquisitions needs to be considered in the detection of stress.

Moreover, different Machine Learning (ML) approaches (i.e., Support Vector Machines, Random Forests, Bayesian Networks, K-Nearest Neighbors and Deep Neural Network) are used in the recognition of stress [46], comfort [49], fatigue or to evaluate the general physical condition [50]. The ML models published in the literature are mainly based on the elaboration signals such as Electroencephalography (EEG) or Electrocardiogram (ECG) [51] signals, galvanic skin response [52] or derived features such as heart rate variability (HRV) [53]. In particular, the ECG signal is widely adopted for estimating levels of stress using a Convolutional Neural Network (CNN), reporting a classification accuracy over 90% when considering signals from relatively stationary subjects [54].

## 3. Monitoring System

The RT-PROFASY system manages and integrates different types of data coming from various sources in order to enable the comprehensive monitoring of all parameters that impact productivity and workplace safety (i.e., human factors, environmental conditions, production processes). Data collection occurs during the execution of the production workflows and is mainly automated. Manual data entry is limited to tracking the start and end of processing steps, the outcome of final checks on products, and the reporting of anomalies that cannot be automatically detected by the system. Input data are associated with the timestamp, the identity and the location of the generating entity to enable spatial and temporal correlation analysis. Raw input data are further elaborated both to obtain complex aggregate data (e.g., overall resource consumption of a process) and to flag potentially dangerous situations through timely notifications.

All collected data can be grouped into categories according to the monitoring purpose to which they relate: work-related stress and physical conditions (data type “S”), production processes, environment and assets localization (data type “P”), and product quality (data type “Q”). A detailed description of such parameters and data sources is reported in Table 1.

The heterogenous dataset provided by the RT-PROFASY system allows analyzing and identifying the causes and taking corrective actions in case of inefficient production processes, logistics or unsatisfactory quality levels. The system also contributes to improving safety by actively monitoring the physical conditions of workers and PPE compliance at each workstation.

The architecture of the monitoring system is shown in Figure 1. The overall system is composed of several subsystems, further detailed in the following subsections, interconnected through an advanced middleware layer that features also data elaboration and storage capabilities. Each subsystem is dedicated to the monitoring of specific parameters and participates with the collected data in the construction of the multidimensional dataset hosted in the central repository.

In particular, the middleware provides a Representational State Transfer (REST) Application Programming Interface (API) based on the micro-services paradigm to enable the subsystems to upload captured data and query the repository. It also manages both intra- and inter-subsystems data exchanges. HTTPS protocol and token-based authentication are enforced for security reasons. All incoming and outcoming data are in JSON format.

In addition to receiving, storing, and retrieving data, the middleware generates derived data and sends unicast, multicast, or broadcast notifications and actuation commands, which may arise from incoming data elaboration. This processing is based on ad hoc developed routines and algorithms, including complex ones such as Machine Learning models, which are associated with the various input data streams. For example, an end-of-process incoming message triggers the calculation of the duration and resource consumption of the entire process, whereas specific notifications addressed to the target workers are produced in case of the arrival of an alarm message from the machinery they are working on or in case acute stress conditions are detected after biomedical data elaboration.

Finally, the middleware provides authenticated users with a flexible web-based front-end interface. It allows the interaction with the RT-PROFASY system by means of customizable dashboards for data browsing, reports generation, and the listing of the required resources to carry out manufacturing processes, production batches, and working activities. Specifically, it is possible to build custom views with tabular or graphical representations of the data in the repository, even combining data from different subsystems and aggregating them on a spatial, temporal or process basis. This represents a useful tool to investigate correlations among S, P and Q parameters and to identify corrective actions needed to improve business productivity.

### 3.1. Biomedical Subsystem

The biomedical subsystem manages the acquisition of body signals and other vital parameters belonging to the S category for fatigue/stress and well-being detection. Vital parameters are collected using a shared Point-of-Care (PoC) installed in the factory, whereas the acquisition of body signals is completed in real time using a wearable device and a smartphone, which runs an ad hoc developed software application (App). All data collected are automatically sent to the middleware exploiting the dedicated API endpoints.

Concerning the real-time monitoring of body signals, the choice of the wearable device is significantly influenced by the considered types of work, especially in the case of manufacturing activities that require specific PPE usage or more in general are characterized by a high operator motion. This leads to discarding solutions such as wristbands and smartwatches due to the high noise affecting the signals, which is caused by the movement of the limbs during work activity; in addition, they could represent a hindrance for PPE such as gloves, thus introducing risks from their use. Instead, the selected sensorized T-shirt, being perfectly adherent to the body of the user, provides good-quality signals even in the presence of movements. In addition, it can be easily worn under the work uniform without impacting worker’s mobility and safety.

The sensorized T-shirt is a COTS device produced by Smartex srl [55]. It is provided with a Software Development Kit (SDK) to facilitate the integration in custom development. The device measures the ECG signal related to the first lead (right arm—left arm) by means of two dry electrodes, the extension of the ribcage during breathing by means of a piezoelectric band, and finally the movement by means of a 9-axis Inertial Measurement Unit (IMU) platform (accelerometers, gyroscopes and magnetometers). In addition to the embedded acquisition electronics, the device is equipped with Bluetooth connectivity for data exchange and a rechargeable battery capable of powering the circuit for the entire work shift. The on-board processing, based on an Atmel microcontroller, also provides derivative signals such as heart and respiratory rate, which can be optionally requested by the user. The complete list of data acquirable from the device is shown in Table 2. Among the supported signals, only the ECG is currently used for stress detection, while the rest of the data are collected to enable further offline analysis and future extensions such as fall or activity detection.

The sensorized T-shirt needs to be paired to the smartphone used for data acquisition and notification management. The chosen smartphone is a commercial rugged device powered by an Android 9 operating system. It features wireless connectivity (i.e., Bluetooth, Wi-Fi and mobile broadband) and storage capability required for the monitoring activity handled by the App.

The App has a threefold objective:To handle data acquisition from the T-shirt;To act as connecting node toward the middleware, to send acquired data and receive messages, alerts, or commands;To present the user with the abovementioned notifications and/or requests coming from local data processing.

The App requires a one-time initial configuration to obtain the identification code (ID) that will uniquely identify the App instance (and therefore the smartphone and the worker to whom the device is assigned) within the RT-PROFASY system. This identifier will be later associated with all the generated data streams and also exploited to address the notification messages to the specific smartphone. As far as the T-shirt pairing procedure is concerned, the App automatically recognizes it among the Bluetooth devices associated with the smartphone and sets it as the active data source. If none or more devices are found, special messages inform the user about the steps to follow for completing the installation. At the end of the configuration and every time the App is launched, the main screen is shown, from which the user can access all the functionalities and start the monitoring.

The monitoring can be carried out in two ways: either in a continuous and automatic manner or when manually requested by the user. In the continuous monitoring mode, acquisitions are started by a timer at fixed intervals, which are defined by customizable settings (by default 60-second-long acquisition every 30 min). Then, there is a button in the App to immediately start data acquisition if an automated one is not in progress. In either case, the App is able to keep the monitoring running even when the App is in the background. This allows the users to lock the smartphone and keep it aside while working. To stop the monitoring, regardless of the chosen mode, a dedicated button is present on the App’s main screen.

In order to minimize the actions requested to the user, the App automatically performs data forwarding to the middleware and manages temporarily Bluetooth disconnections from the T-Shirt, occurring for example in case the worker moves too far away from the smartphone. The only interaction that could be optionally requested to the user, yet not mandatory or blocking, is to provide feedback about the perceived stress level and ongoing working activity at the end of acquisitions. Moreover, to improve robustness and avoid data loss in case of network failures, all incoming data are temporarily stored in the permanent memory of the smartphone and deleted only after a positive transmission acknowledgment is received from the middleware.

The App is also able to receive two types of messages from the middleware (i.e., push notifications), respectively, to convey information resulting from the data processing (e.g., detection of too high stress levels, presence in an interdicted zone, evacuation order, etc.) and to provide additional features such as requesting log or unsent data or changing App settings remotely. The former is presented to the user in the notification bar and as a *Toast* within the App, showing the text in the message. They are also stored and presented in a history list for later reference. The latter is completely transparent to the user (i.e., no visible notification is produced). Moreover, visible notifications are also directly produced by the App as a result of local data processing and to require user attention in case of feedback requests or unrecoverable errors.

Finally, a set of user-modifiable settings is made available on a dedicated App screen. These settings include the duration of the acquisitions and the interval between them in continuous monitoring mode, the activation of the perceived stress request to the user, and the selection of the types of data to acquire among those provided by the T-shirt. Furthermore, information related to the specific installation (i.e., device ID, sensor name, and software version) is displayed. Figure 2 shows the App’s screens: customizable settings (A), main view (B) and feedback request (C).

The aforementioned functionalities are implemented using the Android SDK API level 28, which is compatible with devices running operating systems version 9 or higher. Figure 3 shows the architecture of the implemented App.

The schema presents, with reference to the typical Android SDK terminology, the main functional blocks of the application; data interchange modes among them are also highlighted. To lighten the notation, processing flows for notifications internally generated or received from the RT-PROFASY system are not shown. In particular, gray blocks represent the device resources used by the App for interacting with the user (touch screen), scheduling automatic acquisitions (alarm manager), saving data, and communicating with external systems (file system and wireless resources, respectively). The main front-end is implemented by a single multi-fragment *Activity*, which includes graphical elements for presenting status information and messages, and the buttons to perform actions (e.g., start/end automatic or voluntary acquisition, provide stress feedback, access settings, consult notification history, etc.). Its content changes according to the App current status. Data acquisition, local storage, and transmission are carried out by a *Foreground Service* and two threads, which are dedicated to communication with the T-shirt and the middleware, respectively. Such a service remains active even when the App loses focus or the smartphone is locked. It displays a permanent notification and is destroyed only when the App is intentionally closed. Device permanent storage is used to write incoming data during acquisitions and to read data to be sent during transmissions for preventing data loss in case of errors or connectivity faults. A *SharedPreferences* component stores configuration parameters that are only writable from the configuration screen but are available for reading to all other components.

Push notifications are implemented by exploiting Firebase Cloud Messaging (FCM) provided by Google. It acts as a message proxy between the App and the RT-PROFASY middleware and requires the integration of a dedicated software module both in the sending and receiving component. FCM supports unicast, multicast, and broadcast message delivery. In addition, to prevent unauthorized operations, the parts involved must share a credential file released by Firebase itself. Firebase exploits a generated *token* to address notification messages, which is an alphanumeric string identifying a specific device. Once the token is received by the App (at start-up), it is sent to the middleware together with the device ID (the identifier inserted at installation time). In this way, whenever the middleware has to send a message to a specific App instance, it addresses the notification to the token looked up by device ID.

Finally, in addition to the real-time monitoring with the wearable device described above, the biomedical subsystem includes a Point-of-Care (PoC) module for the acquisition of other vital parameters on a voluntary basis. The PoC consists of an Android 9 tablet and a set of commercial Bluetooth biomedical devices. Through a dedicated App, the PoC allows the workers to take measurements of vital parameters such as blood pressure, temperature, oximetry, and weight. These optional data can contribute to the assessment of the health and well-being of workers but are not used for stress detection. The software application requires the worker to log in with a personal username and password. Once signed in, the user can select the desired measurements and follow the guided procedure to perform the acquisitions. Data acquisition and forwarding to the middleware are handled transparently to the user, which is finally informed of the outcome of the operation.

### 3.2. Machine Learning

The detection of acute stress conditions is carried out by a binary classification algorithm based on Machine Learning (ML), which uses as input the ECG signals acquired during working activities. Conversely, periodic and chronic stress conditions are highlighted in case acute stress episodes are continuously observed over time.

The ECG signal represents the electrical activity of the heart, and its main components are shown in Figure 4. In particular, the QRS complex reflects the electrical activity within the heart during the ventricular contraction; hence, it serves as the basis for classification schemes of the cardiac cycle [56]. Typically, the QRS complex band ranges from 3 to 40 Hz [57], the P-wave ranges from 0.5 to 10 Hz, and the T-wave band ranges from 1 to 20 Hz.

The developed algorithm has to deal with the noise affecting the ECG signals, which is mainly introduced by worker movements and muscle activity. The movement produces artifacts in the measured signal mainly in the 0–5 Hz spectral range [58] due to the sliding of the electrodes over the subject’s body. Whereas, muscle noise is predominant in the band above 35 Hz [59].

Thus, to make usable the acquired ECG signals, a high-pass filter with a cut-out frequency of 5 Hz and a low-pass filter with a cut-out frequency of 35 Hz are used. This results in a strong attenuation of the noise components while keeping the contribution of the QRS complexes almost unchanged. Given the non-diagnostic nature of the system, the limited information loss due to the filtering (i.e., T and P waves) can be endured to acquire a cleaner signal. A comparison of the raw ECG signal acquired under motion conditions and its final filtered version is shown in Figure 5.

In addition, since noise does not affect the signal with the same intensity throughout its duration, a windowing strategy is applied to further reduce the impact of noise on the final classification. The filtered signal is divided into 12-second-long non-overlapping slices, and classification is computed for each of them using a 1D Convolutional Neural Network (CNN). The final classification outcome is elaborated by a majority voting logic in the last stage of the algorithm. In this way, the decision taken on a very noisy segment can be counterbalanced by decisions taken on clearer signal slices, improving the overall performance. Figure 6 shows the block diagram of the stress detection algorithm, where filtering and classification steps are highlighted.

The structure of the CNN is shown in Figure 7. In order to maintain the figure clearly, and since the central layers repeat themselves, the blocks inside the box appear in the actual network six times. The CNN presents 25 one-dimensional convolutional layers with a filter size of 9, which are followed by a dense layer with 1024 nodes. Batch normalizations and Rectified Linear Unit (ReLu) activation functions are used after each convolutional layer. Every two convolutional layers, apart from the first one, a skip connection connects the n-th layer with the n+2-th layer. In order to progressively reduce the input size, every four convolutional layers, a 1D MaxPooling with stride = 2 is applied, and after each skip connection, the next layer also has a value of strides = 2 to compensate for the increase of input size caused by the connection itself. The final dense layer is preceded by a Global Maxpooling and uses a Softmax activation function and dropout normalization.

The supervised training of the CNN was completed in two distinct phases, which are called pretext and downstream task, respectively. The pretext task allowed the network to learn robust generalized features from ECG signals (i.e., high-level abstract representations and recognition of signal transformations) [60]. In the second task, the network was trained to perform the required stress classification.

During the pretext task, the network learnt ECG spatiotemporal representations by recognizing four different types of ECG signal transformations: unchanged, inverted in time, inverted in space, and inverted in both time and space. The dataset used for this scope is Physionet [61], which provides long-duration ECG traces sampled at 250 Hz concerning 11,000 patients. The original signals were divided into 2-min portions; then, time and space transformations were applied. Out of the whole resulting dataset, 75% was used for training, 15% was used for validation, and 10% was used for testing. The model reaches an accuracy of 98.4% for the pretext classification task evaluated on the testing dataset.

The downstream task trained the network to perform stress classification. It used as its starting point the trained network resulting from the pretext task. According to the transfer learning approach [62], the obtained weights of the convolutional layers were transferred to a second identical network, and the final dense layer was replaced by another one of the same type. This dense layer is the only one trained in this task. To this aim, the dataset was built using ECG signals coming from databases publicly available [63,64], a database available for research purposes through access request [65], and signals taken on the field during a preliminary acquisition campaign performed with the RT-PROFASY system itself. In all cases, each trace is labeled with the corresponding perceived stress level.

As reported in Table 3, the final downstream dataset is composed of 93% ECG signals coming from the literature, and 34% of the total is labeled as signals of stressed people. In the case of the signals collected on the field, it was the workers themselves who labeled the traces through the App, according to the perceived stress. As before, the dataset was divided into three parts: training (75%), validation(15%), and testing (10%).

The final network gained an accuracy of 88.4%, with a F1-score of 0.90 in the binary classification of stress. The model is implemented using the TensorFlow library and has a footprint of 25 MB. Since TensorFlow comes also in a Lite version, it will be possible in the future to integrate the model directly into edge devices (i.e., smartphones), so that stress levels’ classification could be completed without sending significant amounts of data to the middleware.

### 3.3. Building Automation Subsystem

The P data type collection is handled by the Building Automation subsystem, which measures environmental parameters and acquires operational data from the production site’s industrial machines and plants with the aim of environment, process, and safety conditions monitoring.

The system is implemented with Programmable Logic Controllers (PLCs) produced by IEI Integration Corp. and sets of commercial I/O boards and sensors distributed within the factory. The backbone interconnection uses an industrial RS485 bus. In order to increase fault tolerance and implement recovery procedures, the subsystem architecture exploits redundancy and distributed components. It is composed of the Centralized Master PLC (CM-PLC), the Centralized Backup PLC (CB-PLC), and several Local Backup PLC (LB-PLC), which are placed in proximity of the machinery to be controlled. CM-PLC and CB-PLC are embedded Debian systems featuring specialized hardware and running custom software applications implemented for specific control and communication tasks within the RT-PROFASY system. They both have two RJ45 connectors: one is toward the local network (LAN) where the middleware is hosted to enable bidirectional communication and PLC remote configuration; the other is connected directly to the Ethernet RS485 bus bridge communication board (BCB) to exchange data with the distributed nodes connected to the backbone bus. This bridge automatically switches to the CB-PLC in case the connection with the CM-PLC fails, loading the latest setup and parameters. Finally, several LB-PLCs (Raspberry Pi3 boards with a light version of the embedded Debian system and ad hoc developed software) are located close to each machinery to manage the acquisition and sending of data gathered by the local I/O board. Each I/O board has wired connections to the BCB through a double-wired bus RS485 with proprietary FB NET protocol and with the sensors and actuators installed on the machinery.

To the aim of environmental monitoring, the integrated sensors’ and actuators’ sets provide worker presence detection (using InfraRed technology) and measurements of temperature, brightness, and humidity. All installed sensors are identified with a unique ID, which allows the environmental data to be further correlated with stress, well-being, and process performance analysis. Moreover, automatic procedures or manual interventions (using the Workplace UI) allow the adjustment of environmental variables to improve the worker’s comfort.

As far as process control and safety are concerned, the subsystem directly interfaces production machinery and plants. In this way, it is able to acquire instant and overall power consumption estimation of the production processes as well as detect and broadcast warning events triggered, for example, in case of raw materials shortage or operative anomalies (i.e., level below the specific threshold). Nevertheless, corrective actions are not automated and must be performed directly on the involved machine by human operators. In addition, each workstation is provided with dedicated buttons to signal dangerous situations which, when pressed, produce notifications sent to PLCs and then to other workstations and users via middleware.

More in general, data exchange within the building automation subsystem and toward the RT-PROFASY middleware happens either at fixed sampling rates (for analog quantities whose dynamics change slowly, i.e., temperature, humidity and brightness sensors, power consumption) or as soon as the information/event is detected (for digital quantities, e.g., presence sensors, triggers, warning events, changes of state, etc.).

Finally, some post-processing optimization criteria are implemented in the software routines to balance the quantity of data logged and transmitted with respect to the usefulness of the data collected as well as save storage space and speed up research and analysis.

### 3.4. RTLS Subsystem

The tracking of objects, assets, and the PPE’s position within the production site is managed by the Real-Time Location System (RTLS) subsystem. The availability of positioning data with sub-meter precision and, at the same time, the virtual segmentation of factory spaces into areas, allowed the implementation of software routines to:Verify PPE compliance with respect to area-specific rules;Verify accesses made to workstations and restricted areas;Search for work tools, product batches, semi-finished products, and raw materials;Stand by machinery in case of an unattended workstation.

This subsystem is composed of COTS hardware resources produced by Ubisense: a centralized RTLS server, several receiver antennas placed inside the factory at specific coordinates (called references), and active tags placed onto the items to be localized. The smart antennas compute the position of each tag by measuring Time Distance of Arrival (TDoA), Angle of Arrival (AoA), and the strength of the signal (see Figure 8). The localization process happens in real time, and all the positioning data are processed by the RTLS server and immediately made available to the RT-PROFASY middleware.

The RTLS subsystem uses Ultra-Wideband (UWB) technology with a signal strength of 2.4 GHz, which allows for sub-meter accuracy in real time (less than 1 s) and potential scalability over 1000 tags. In addition, the use of UWB technology enables the possibility of deriving the pin-point 3D location, which helps in the detection of dynamic equipment movements, and the identification of people/assets in critical processes. An alternative technology that could be adopted is based on Bluetooth, whose hardware is less expensive but provides worse performance, especially in 3D positioning.

Given the industrial environment, in which signals can be reflected and/or obstructed, antennas may measure incorrect positioning and/or physically impossible movements. In order to detect and remove these outliers, the RTLS subsystem applies advanced Kalman-based filtering algorithms, which distinguish the direct signal path from its reflections.

Overall, the availability of precise positioning data enables the middleware to raise alarms and send notifications to the workers whenever they enter a restricted area or are not wearing the proper PPE, contributing to the prevention of dangerous situations. In addition, it enables the analysis of the correlation between stressful situations and the specific site location, while production times could benefit from the search for assets capability offered by the subsystem.

### 3.5. Workplace UI Subsystem

The Workplace User Interface (UI) allows the interaction between the worker and the RT-PROFASY system. It is an ad hoc implemented web-based UI (Figure 9) made available in the production site through different types of hardware resources, the choice of which needs to keep both workers and machinery safe. For example, some machines have an industrial Linux-embedded 10” touchscreen panel placed nearby, but when this is not possible, workers can interact with the UI through the same smartphone used for the Biomedical subsystem and/or a shared 10” industrial rugged Android OS tablet. In either case, the connection to the middleware exploits the Wi-Fi network available in the factory.

This UI also provides useful information to better carry out the job, for instance showing the list of activities, the bill of materials required, tracking the status of single activities, and entire processes by recording their start/end time, etc. It also allows reporting problems that occur during production activities, including interruptions or discarded products. Overall, the information contributes to the “P” data type by monitoring the evolution and outcomes of processes and coordinating subsequent production phases.

In addition, this subsystem optionally allows the visualization of the environmental parameters measured in the proximity of the workstation and, where available, to modify them by acting on the actuators of the Building Automation subsystem.

For the purposes of safety, through the Workplace UI, the workers could receive immediate notification about dangers/fail machinery events raised by the local machine or elsewhere on the production site. Similarly, they can signal emergencies to all their colleagues.

Finally, this subsystem provides functionalities related to RTLS components such as:Search for the location of PPE required to operate the associated machine or enter a specific area;Search for the location of assets and materials needed to accomplish the assigned activity;Receive timely notification if PPE is missing.

## 4. On-Field Trial

An on-field test of the RT-PROFASY system was conducted in order to assess the technological and operational feasibility, the achieved performance, and the impact on productivity, production quality, and workers’ perceived working conditions in a real application scenario.

The selected testing site is a private company operating in the mechanical manufacturing and metalworking sectors. It is mainly focused on metal heat treatment and metal carpentry. The production is largely characterized by custom products, non-repeatable items, and small production batches. The company also offers complementary services such as technological tests and precision measurements. Two main different categories of work activities carried out in the production site results from the preliminary analysis:Activities with high risk and fairly slow dynamics (few pieces produced during the day) typical of heat treatment of metals;Activities with faster dynamics but characterized by repetitive actions, such as the production of several identical pieces, apparently having a lower risk level (i.e., metal carpentry).

In all cases, the work activities are predominantly physical and characterized by a high level of movement of the operator. It is worth noting that activities with similar characteristics can also be found in other manufacturing sectors, allowing the developed system to be applied in other industrial contexts.

The setup of the prototype system included the installation of the Building Automation and RTLS subsystems components and their interconnection to the middleware. The latter was deployed on a dedicated server machine in the cloud. A total of seven pieces of production machinery were connected to the infrastructure for reading operating parameters. In addition, each workplace was equipped with sensors to monitor presence and environmental conditions and the UI to allow the worker to interact with the system. Eight UWB-based antennas were deployed in the area, and location tags were applied to the main PPE and equipment used during production. Moreover, the factory space was virtually divided into 10 sectors, defining the PPE compliance rules for each one. WiFi connectivity was extended to cover the entire area of the production site.

A total of 10 workers were involved in the test over a period of 4 months. Informed consent was obtained from all subjects involved in the study. All of them attended an initial training session (2 h) and received personal monitoring devices already configured (i.e., T-shirt and smartphone). They were also informed of the availability of additional training as needed. No user interrupted the test before the end.

No accidents occurred during the trial, nor were there any observed events or behaviors ascribable to periodic or chronic stress conditions. The absence of such forms of stress was confirmed by the analysis of stress levels calculated by the system and also using traditional evaluations carried out at the end of the trial by the company doctor. Data analysis also showed high PPE compliance in the factory.

A few episodes of acute stress and fatigue were detected, especially while performing particularly physically demanding activities for medium/long periods and on certain days when the ambient temperature detected at the workplace was very low (minimum measured 11 ${}^{\circ }$C). Such episodes were sporadic and had a limited duration. They were always reported to the worker via a dedicated notification on the smartphone and did not cause safety issues nor affect the quality of production in terms of human errors. Specifically, the ML model developed for detecting acute stress from ECG signals acquired in motion features an accuracy of 88.4% and an F1-score of 0.90. This performance is comparable with models in the literature that use signals acquired under stationary conditions of the subject.

The impact on the production quality was evaluated by comparing the number of defective products, waste, and manufacturing errors occurred during the trial period vs. the ones that occurred in the same production processes in previous periods. To this aim, past data provided by the company were used. From this point of view, there was no deviation in performance. In addition, the duration of individual activities and entire processes proved to be in line with the past. It should be noted that the company has long applied policies to minimize errors and organized processes to maximize performance and product quality.

The standardized System Usability Scale (SUS) was used to measure the usability perception of the RT-PROFASY system. The SUS consists of ten questions, which are positively and negatively oriented. The response scale for each question is a 5-point agreement scale from strongly disagree (1) to strongly agree (5). Workers were asked to answer the survey in anonymous mode. The scores of each question were normalized in the 0 to 4 interval by subtracting 1 from the original score of positive-oriented questions and by subtracting the original score from 5 for negative-oriented questions. For each survey, the SUS 0–100 score was obtained by summing the normalized scores of all questions and multiplying that sum by 2.5. The good usability level of the system is confirmed by the average final SUS score of 73 (grade B), which converts to a percentile rank near 70%. The ease of use of the system is also confirmed by the fact that no worker required additional training beyond the initial session and was able to operate the system without any particular problems.

The impact on safety, organization of work, and in general on the quality-of-life during activities perceived by workers was evaluated through a dedicated survey filled in anonymously. The five positive-oriented questions included a 5-point agreement response scale as for the SUS, and the same normalization approach was used. Table 4 shows the average score obtained for each question. All questions received an average rating from sufficient to positive, among which the perception of health protection stands out. In addition, all workers confirmed that the T-shirt did not interfere with work activities and was a good solution in terms of usability and ergonomics. The main disadvantage pointed out was the need for frequent washing of the fabric. However, no particular comfort or hygiene issues due to prolonged use of the T-shirt were reported.

The robustness and reliability of the RT-PROFASY system from a technical point of view were indirectly assessed through the monitoring of malfunctions and requests for maintenance intervention during the trial. No major malfunctions negatively interfered with production, while temporary malfunctions (i.e., lack of connectivity or power supply, etc.) at some nodes were automatically recovered without causing any data loss. All interventions carried out concerned the improvement and optimization of hardware and software resources to refine data acquisition and processing capabilities.

Ultimately, the RT-PROFASY system demonstrated its technical and operational feasibility and achieved the expected performance. The real-time monitoring of all factors affecting productivity, feasible through the availability of a solid and constantly updated heterogeneous dataset enables the identification of potential problems and improves the working and environmental conditions, the safety of workers, and definitely the quality of the production.

## 5. Conclusions

This paper describes a complete system aimed at monitoring in an integrated and reliable way all factors that affect both the well-being and safety of workers and business productivity. The system is composed of COTS hardware components and custom-developed software applications and routines to enable real-time data acquisition and processing throughout the execution of the production workflows. The system supports different kinds of data related to production processes and machinery, environmental conditions, and human factors (i.e., stress and fatigue), which altogether contribute to productivity and the quality of production.

The availability of such a multidimensional and comprehensive dataset allows detailed analysis of the performance of production processes taking into account multiple aspects, also enabling the identification of the causes of possible worsening and the measurement of impact of corrective actions. Beyond the optimization of the working activities, it also aims at improving safety conditions and ultimately pursuing production quality through sustainable processes for workers and appropriate working conditions.

Future work concerns the investigation of the impact of collected biosignals other than ECG for the stress detection task and the exploitation of inertial data for the implementation of new features such as fall detection and activity recognition. Furthermore, the implementation of the current stress detection algorithm on edge devices and the study of an extended ML-based classification model providing a stress level score instead of a binary output are foreseen.

## Figures and Tables

**Figure 1 sensors-23-05502-f001:**
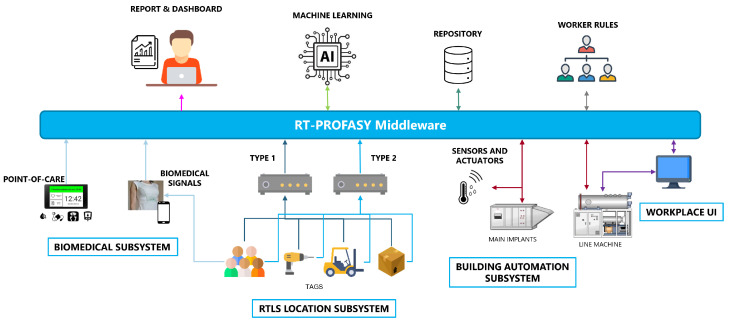
Architecture of the RT-PROFASY monitoring system.

**Figure 2 sensors-23-05502-f002:**
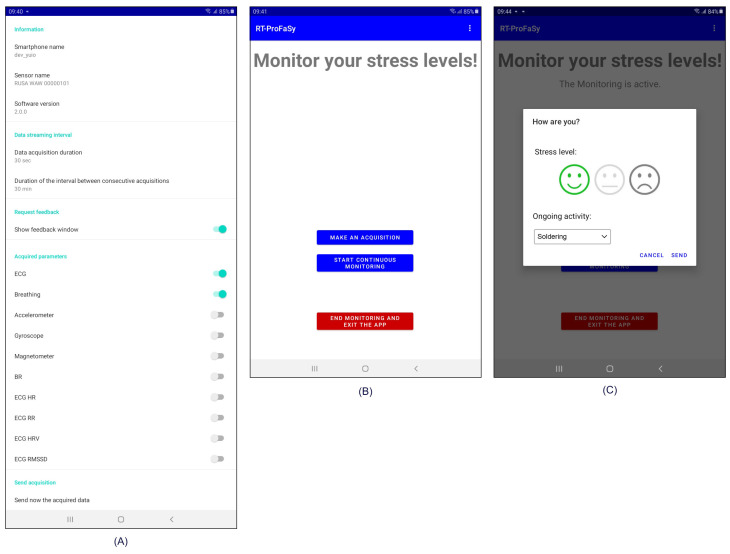
App’s main screens: customizable settings (**A**), main view (**B**) and feedback request (**C**).

**Figure 3 sensors-23-05502-f003:**
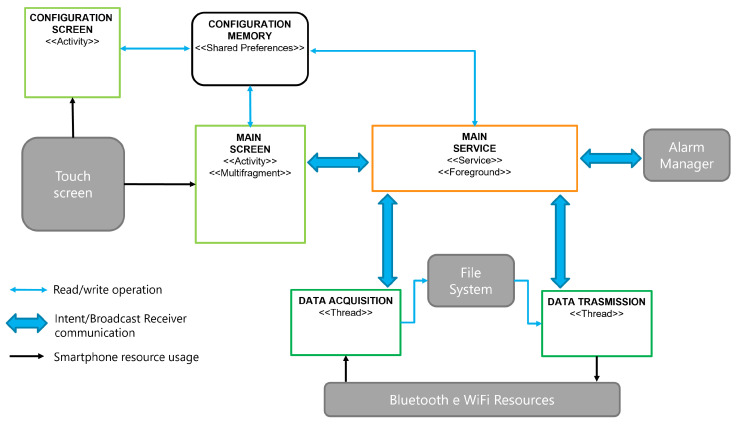
Architecture of the RT-PROFASY Android application.

**Figure 4 sensors-23-05502-f004:**
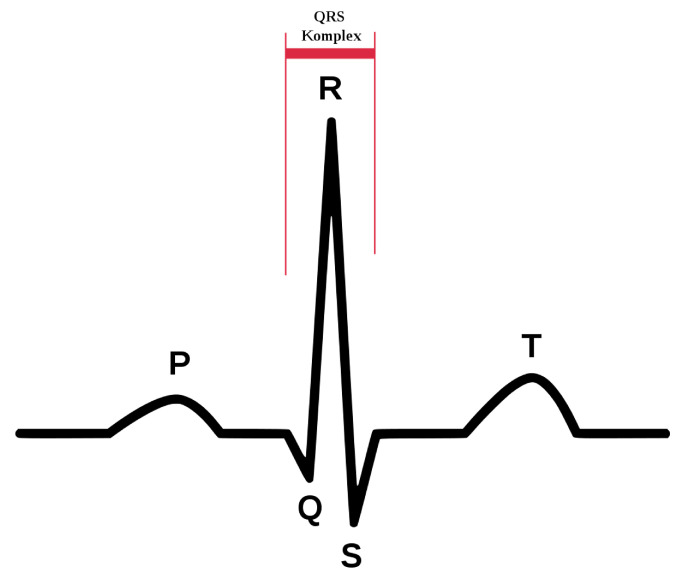
ECG signal main components: QRS complex, P wave and T wave.

**Figure 5 sensors-23-05502-f005:**
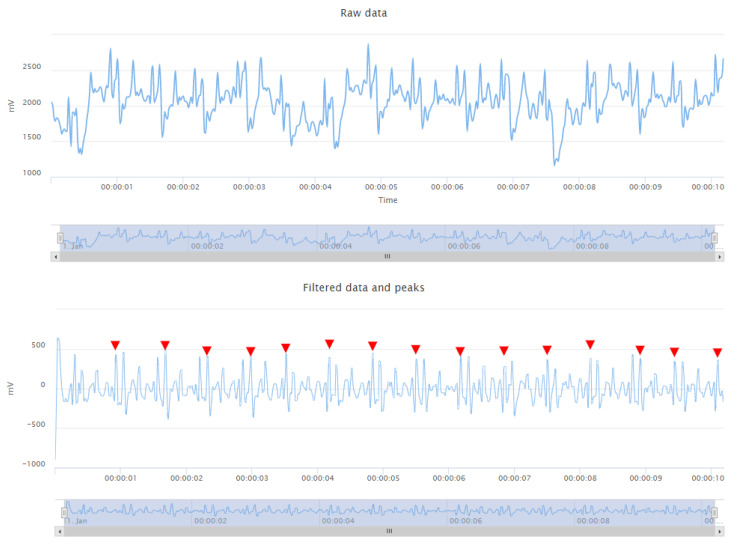
Comparison of raw (**top**) and filtered (**bottom**) signal under moderate motion conditions.

**Figure 6 sensors-23-05502-f006:**
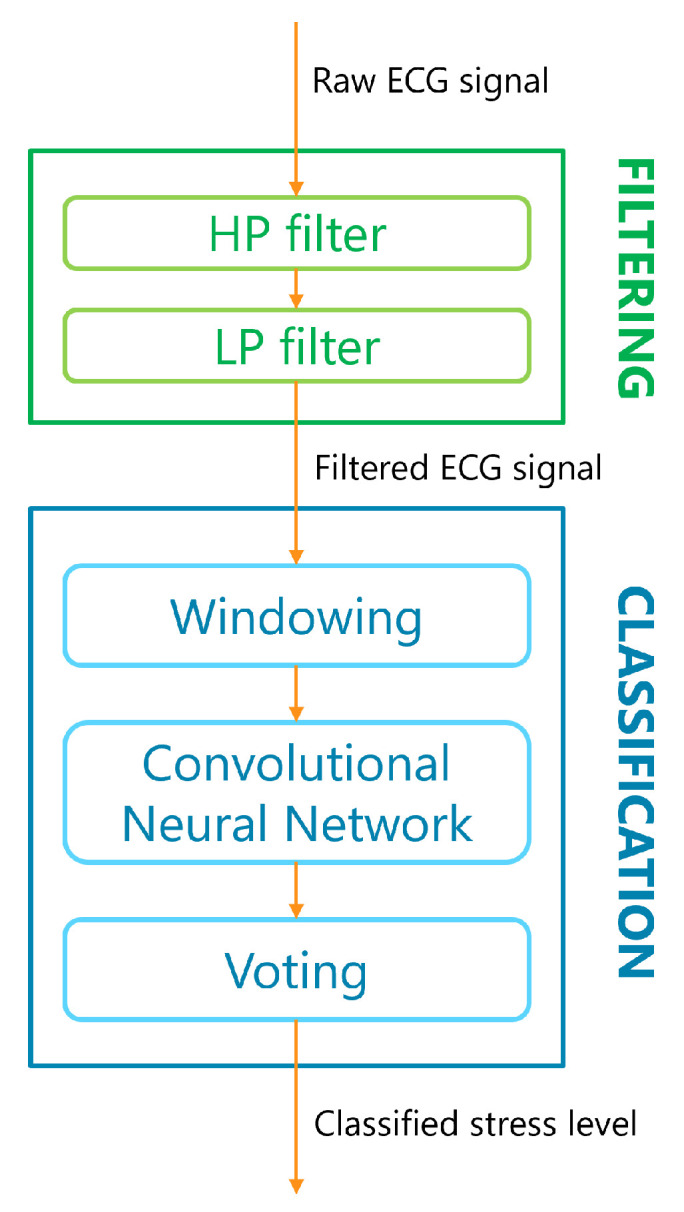
Block diagram of the stress detection algorithm.

**Figure 7 sensors-23-05502-f007:**
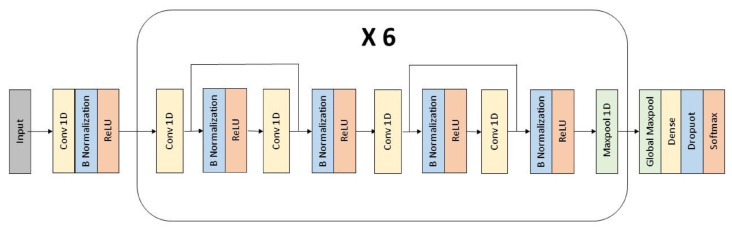
Structure of the CNN for stress classification.

**Figure 8 sensors-23-05502-f008:**
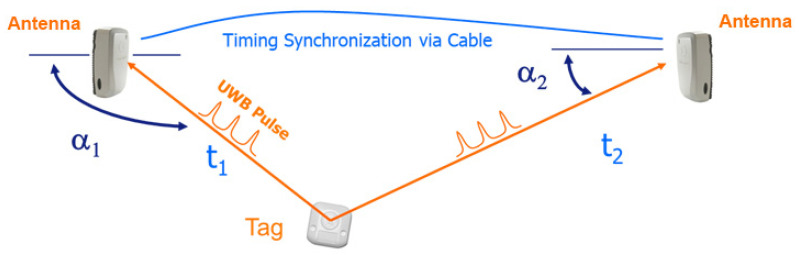
RTLS subsystem: localization computation.

**Figure 9 sensors-23-05502-f009:**
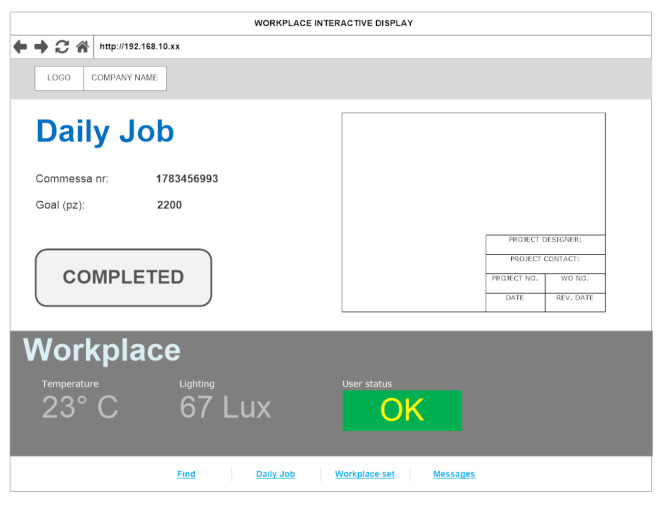
Workplace UI: web-based interface.

**Table 1 sensors-23-05502-t001:** Summary of collected parameters per category.

Category	Monitoring Target	Description	Source Subsystem
**S**	Work-related stress and fatigue condition	Biometrics data: ECG, breath, pulse oximetry, blood pressure, temperature etc.	Biomedical
**P**	Production processes, environment conditions and indoor localization	Process timelines (start, stop, duration etc.); raw materials and semi-products usage; process intermediate and final outcomes and wastes; environmental conditions (temperature, humidity, brightness etc.); assets/PPE localization tracking	Building Automation, Workplace User Interface and Real-Time Location Subsystem
**Q**	Quality of final product	Quality and acceptance checks; inspection, measurements and evaluation of final product vs. requirements and expected result; evaluation of overall production cost and time	Workplace User Interface

**Table 2 sensors-23-05502-t002:** List of raw signals and derivative data supported by the sensorized T-shirt.

Signal Type	Description (Representation, Sampling Frequency)	Raw/Derivative
ECG	Electrocardiogram (12 bit signed, 250 Hz)	Raw
RespPiezo	Breathing from the piezo sensor (12 bit signed, 25 Hz)	Raw
AccX, AccY, AccZ	Acceleration along X, Y and Z axes (12 bit signed, 25 Hz)	Raw
GyroX, GyroY, GyroZ	Gyroscope along X, Y and Z axes (12 bit signed, 25 Hz)	Raw
MagX, MagY, MagZ	Magnitude along X, Y and Z axes (12 bit signed, 25 Hz)	Raw
ECGHR	Heart rate (0.2 Hz)	Derivative
ECGRR	RR peaks distance (4 Hz)	Derivative
ECGHRV	Heart Rate Variability in the time domain (0.01 Hz)	Derivative
ECGRMSSD	RMSSD computed from RR interval (0.01 Hz)	Derivative
BR	Breathing rate (0.06 Hz)	Derivative
BA	Breathing amplitude (0.06 Hz)	Derivative
ActivityPace	Number of steps per minute (0.2 Hz)	Derivative

**Table 3 sensors-23-05502-t003:** Dataset composition: number of 12-second-long segments per stress label and source (literature or on the field).

Label	Number of Literature Signals	Number of on-Field Signals	Total
**Not stressed**	16,090	1740	17,830 (66%)
**Stressed**	9000	250	9250 (34%)
**Total**	25,090 (93%)	1990 (7%)	27,080

**Table 4 sensors-23-05502-t004:** Questions proposed in addition to the SUS survey.

Question	Score [0–4]
The use of the system made my work organization easier	2.4
The use of the system made my work safer	2.2
The use of the system improved the environmental working conditions	2.4
I think that using the system, my health status is more protected	2.8
The use of the system improved the quality of my workday	2.2

## Data Availability

Not applicable.

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
