# Peer review of "ECG-Based Stress Detection and Productivity Factors Monitoring: The Real-Time Production Factory System"

_sensors, 2023, doi:10.3390/s23125502_

Round 1

Reviewer 1 Report

Even though the language is of a high quality, the paper is written in a particularly verbose mode which makes it irritating to read.

It is completely not clear what is developed in terms of software and hardware and what is just a concept presented. For instance, is the t-shirt a new or an existing one; is the general system description just a concept or that system really exists? If things have been developed one needs to say who developed them.

The paper is full of irrelevant information and inconsistencies, which makes impossible for me to figure out what was really done. A crucial part in detecting stress in this paper is training neural network with ECG. This is however described very unconvincing. The authors say they used a database from the Physionet. The database however has nothing to do with stress, and then they say that they additionally used databases, used by some previous publications, but these databases do not appear to be publicly available. The question remains were and how did they acquired these databases. It makes me suspicios that authors are not telling the truth here.

If it is the case that the authors have really developed a comprehensive factory monitoring system, the paper does not convince me that is the case. I would suggest to the authors to rewrite the paper from the beginning making it drastically more concise and direct in explaining what was developed and how it was tested. In addition, it is crucial to distinguish what is proposal (proposed systems structure), platform, and what is an actual system.

Reviewer 2 Report

A real-time production factory system is proposed to detect workers’ stress and fatigue in real time using wearable sensors and machine learning techniques and also integrates all data regarding the monitoring of production processes and the work environment into a single platform. Some issues need to be addressed before publication.

(1)   Is there a table between Line 144 and Line 145? Why is there no title? And there is an error in Line 174 that “Tab ??”.

(2)   As many body signals can be acquired by the sensorized T-shirt, why only ECG is applied to stress detection?

(3)   The sensorized T-shirt is functional because there are lots of sensors. Furthermore, some sensors need to be tightly attached to the skin to ensure accurate measurement results. Will wear them for a long time cause trouble for workers, such as discomfort and hygiene issues?

(4)   The established CNN model can realize a binary stress classification for workers. But the stress level of workers should be continuous, not a binary state. It may be more reasonable to provide scores that reflect the level of stress.

Reviewer 3 Report

1. All the figures can be improved. The table on Page 4 is not clear.

2. The references should conclude the latest work.

3. Some future topics should be given in conclusion.

The English writting is good.

Round 2

Reviewer 1 Report

I do not understand what is the purpose of training neural network to "unchanged, inverted in time, inverted in space, and inverted in both time and space". If there is a justification for that, it should be written in the paper.

Fig 4 - where it came from.

Why is the QRS the most important segment.

Need to justify band pass filtering from 5 to 35 Hz. This is in contrast with the statement that QRS is the most important because you will loose QRS with this filtering.

No connection between ECG part and the rest of the system.

The paper is by far too verbose and imprecise and very hard to follow. It needs to be rewritten from the scratch.
